# Temporomandibular joint disorders in skeletal class II patients referred to orthognathic surgery: A cross-sectional study

**Thalles Moreira Suassuna**[1], **Allan Vinícius Martins de-Barros**[1], **Bárbara Araújo da Silva**[2]*, **Fernanda Souto Maior dos Santos Araújo**[3☉], **Tatiane Fonseca Faro**[1☉], **Eudes Euler de Souza Lucena**[4], **José Rodrigues Laureano Filho**[2], **Emanuel Dias de Oliveira e Silva**[2], **Fábio Andrey da Costa Araújo**[2]

1 Post Graduation program of School of Dentistry, University of Pernambuco, Recife, Brazil, 2 Department of Oral and Maxillofacial Surgery, School of Dentistry, University of Pernambuco, Recife, Brazil, 3 School of Dentistry, University of Pernambuco, Recife, Brazil, 4 Multicampi Medical Science School, Federal University of Rio Grande do Norte, Natal, Brazil

☉ These authors contributed equally to this work.
* barbara.araujosilva@upe.br

**Data Availability Statement:** All relevant data are within the manuscript and its Supporting information files.

## Abstract

Objectives: The aim of this study was to assess the prevalence of Temporomandibular Disorders (TMD) in subjects with skeletal class II dentofacial deformity referred for orthognathic surgery, as well as to elucidate its association with sociodemographic and psychosocial features. Methods: This was a cross-sectional study using the Research Diagnostic Criteria for Temporomandibular Disorders. The sample comprised class II skeletal patients referred to an Oral and Maxillofacial Surgery center in the Brazilian Northeast. Results: Seventy-three subjects were enrolled and completed the data collection, which consisted of a physical examination according to Axis I of the Research Diagnostic Criteria for Temporomandibular Disorders and facial analysis. Women represented 82.2% of the sample. Among the assessed subjects, 68.5% were already undergoing orthodontic treatment, and the mean overjet of patients was 6.97 mm. The prevalence of TMD in this sample was 46.6%, with muscular disorders being the most common. Patients with an anteroposterior discrepancy greater than 7 mm showed a higher occurrence of TMD (p = 0.017). Conclusion: This study demonstrated a high prevalence of TMD in skeletal class II patients referred for orthognathic surgery, especially in those with a pronounced overjet, being Group I (muscular disorders) and Group III (degenerative disorders) the most prevalent.

## Introduction

Dentofacial deformities are skeletal jaw discrepancies that result from alterations in growth and the development of dental arches and facial bones. These deformities promote changes in orofacial structures, including jaw muscles and temporomandibular joints, which usually lead to occlusal instability and gross malocclusions with functional, aesthetic, and psychosocial impairment [1, 2]. The Class II deformity is the most common malocclusion pattern, and the

**Funding:** The author(s) received no specific funding for this work.

**Competing interests:** The authors have declared that no competing interests exist.

skeletal features thereof are characterized by anteroposterior disharmony with underdevelopment of the mandible and/or maxillary prognathism [3]. Despite the low quality of the available literature on this topic, in a systematic review Manfredini et al. [4] suggests that skeletal Class II profiles are likely associated with an increased frequency of temporomandibular disorders.

Temporomandibular disorders (TMD) are a heterogeneous group of clinical conditions affecting masticatory musculature, temporomandibular joint (TMJ) or both, and are recognized as the most common chronic orofacial pain of nondental origin [4, 5]. The etiology of TMD is multifactorial, with a wide variety of predisposing and causal factors, such as anxiety, depression, parafuncional activities, trauma, severe pain stimuli, vitamin D deficiency and posture problems [6–9]. While the role of occlusion in the beginning or perpetuation of TMD remains controversial [6, 7, 10], studies suggest that subjects with malocclusions have a significantly higher prevalence of signs and symptoms of TMD than others [1, 11].

Currently, a wide range of conservative treatments for temporomandibular dysfunctions, supported by robust scientific evidence have been brought to light. Among these approaches, manual therapy, therapeutic exercise-based programs, photobiomodulation, splinting, acupuncture, and even the application of botulinum toxin and hyaluronic acid through infiltration have yielded good results in the treatment of TMD [12, 13].

Multiple factors may be associated with the risk of TMD in patients referred for orthognathic surgery, including morphological, demographic, and psychosocial alterations resulting from dentofacial deformities [14, 15]. However, there are few studies assessing whether individuals with skeletal Class II dentofacial deformity referred for orthognathic surgery have higher prevalence of TMD [10].

Therefore, the aim of this study was to assess the prevalence of TMD in subjects with skeletal Class II dentofacial deformity referred for orthognathic surgery, as well as to elucidate the association thereof with sociodemographic and psychosocial features. Our hypothesis is that the prevalence of signs/symptoms and diagnosed TMD according to Research Diagnostic Criteria for Temporomandibular Disorders (RDC/TMD) in this group is higher than in the general population.

A list of all the abbreviations used in this manuscript can be found in the S1 Table.

## Materials and methods

### Study design and ethical considerations

The researchers implemented a cross-sectional study design that was developed in the Northeast of Brazil from March 2014 to November 2016. The study was approved by the Committee of Ethics of the University of Pernambuco, under protocol no. 240/11 (CAAE 23196313.2.000.5207), performed according to the Declaration of Helsinki, and followed the Strengthening the Reporting of Observational Studies in Epidemiology (STROBE) [16] recommendations for cross-sectional studies.

All participants were informed of the research objectives, as well as the risks and benefits, and those who agreed to participate in the study signed an Informed Consent Form.

### Study population

The study population was composed of adult individuals with skeletal Class II malocclusion who were referred for orthognathic surgery. The inclusion criteria of the study were: individuals of both genders at least 18 years of age, diagnosed with skeletal Class II malocclusion in a cephalometric analysis, with indication or under orthosurgical treatment. Subjects with

systemic rheumatic, muscular or joint disorders, history of facial trauma, dental prosthetic space, and cognitive or psychiatric disorders were excluded from the study.

## Sample size

The sample size calculation was performed in the OpenEpi Version 3.01 statistical software using as reference the prevalence of 65% for TMD in the Brazilian population [17]. The confidence interval was set at 95% ($\alpha = 0.05$) with an acceptable margin of error of 10% and a design effect (EDFF) of 1.0. The sample size for an infinite population adjusted for a non-response rate of 10% was defined as 79 subjects.

## Variables assessment and data collection

To reduce the risk of bias and guarantee standardization in data collection, the variables assessment was performed by a single examiner with experience in Oral and Maxillofacial Surgery who were previously trained and calibrated (kappa = 0.83). The variable assessment was composed of a structured questionnaire, facial analysis and TMD examination.

The structured questionnaire was comprised of questions regarding demographics, general health status, parafunctional oral habits, masticatory function and history of orthodontic treatment.

A facial analysis was used to determine the magnitude of dentofacial deformity and was performed according to the principles stated by Arnett and McLaughlin (2004) [18] with a calibrated digital caliper. With the subject's head in a natural position and the labial musculature relaxed, the thyromental distance, which consisted in the distance between the soft pogonion and the superior limit of the thyroid cartilage, and the overjet were measured and registered in millimeters.

The presence and classification of TMD were assessed using the Axis I component of the Research Diagnostic Criteria for TMD (RDC/TMD) [19], which consists of a standardized examination of orofacial structures, and was performed at the dental clinic Department of Oral and Maxillofacial Surgery. Given the specific combination of signs and symptoms gathered in the clinical examination, the subjects were diagnosed and classified according to the criteria in three groups:

GI: muscular disorders (IA—myofascial pain, IB—myofascial pain with limited mouth opening);

GII: joint disorders (IIA—disc displacement with reduction, IIB—disc displacement without reduction, with limited mouth opening, IIC—disc displacement without reduction, without limitation of mouth opening); and

GIII: degenerative disorders (IIIA—arthralgia, IIIB—osteoarthritis, IIIC—chronic non-inflammatory arthrosis).

As the subjects can show more than one condition, the RDC/TMD allows to group them in groups with one or more diagnoses of TMD.

## Database and statistical methods

The database of this study was built on IBM SPSS® Version 17.0 software platform (IBM Corp., Armonk, NY, USA). This process was performed through double tabulation to minimize the risk of typing errors, where two different typists independently typed the same files. The statistical testing phase began upon verification of agreement between the two databases. The variables were dichotomically categorized to enable a bivariate analysis. Which was then

performed through Prevalence Ratios and the chi-square test. A significance level of 5% (p < 0.05) was adopted.

## Results

Of the 79 patients who were enrolled in the study, 73 (92.5%) completed all steps. The only cause of loss for the six patients was the impossibility of completing the RDC/TMD axis I application. This loss did not cause any bias in the results, given that the sample calculation predicted a loss of up to 10%.

The average age of the people who made up the sample was 27.94 years ± 8.73 with a minimum value of 18 years and a maximum of 63. A large part of the sample consisted of women (82.2%), between 18–21 years of age (28.8%), who self-reported as white (80.8%), were undergoing corrective orthodontic treatment as part of the preparation for orthognathic surgery (68.5%), were not in any kind of treatment for TMD in the last 6 months and chewed more on the left side (42.5%), as shown in Table 1. It was still possible to observe that although most of them did not complain about parafunctional activities (53.0%), such activities still registered a considerable number.

The orthodontic treatment average of patients was 30 months ± 34.38, with an average anteroposterior discrepancy of 9.97 mm ± 1.89, a chin-neck distance of approximately 43.65 mm ± 9.21.

Only 34 (47%) of the total number of people examined had a positive diagnosis for TMD; of these, 19 (55%) joined the GI. As it relates to GII, there was a higher prevalence with a diagnosis of left TMJ 11 (32%), and the most associated condition in both was IIA. Regarding GIII, 16 (47%) of the patients were affected by right TMJ, with similar frequencies between the subgroups. It is also worth mentioning that seven (20.5%) were only linked only to GI, two (5.5%) to GII and nine (26.4%) to GIII (Table 2).

**Table 1. Absolute and relative frequencies of subject's demographics and habits variables.**

| Variable | | n | (%) |
|---|---|---|---|
| **Sex** | Men | 13 | (17.8%) |
| | Women | 60 | (82.8%) |
| **Age range** | 18–21 years | 21 | (28.8%) |
| | 22–27 years | 18 | (24.7%) |
| | 28–33 years | 16 | (21.9%) |
| | ≥ 34 years | 18 | (24.7%) |
| **Color** | White | 59 | (80.8%) |
| | Black | 14 | (19.8%) |
| **Orthodontic Treatment** | Yes | 50 | (68.5%) |
| | No | 23 | (31.5%) |
| **Masticatory quality** | Good | 24 | (32.9%) |
| | Bad | 49 | (67.1%) |
| **Masticatory pattern** | Bilateral | 15 | (20.5%) |
| | Unilateral right | 27 | (37.0%) |
| | Unilateral left | 31 | (42.5%) |
| **Presence of parafunction** | Absent | 39 | (46.6%) |
| | Bruxism | 13 | (17.8%) |
| | Tightness | 10 | (13.7%) |
| | Onychophagy | 16 | (21.9%) |
| | Object interposition | 12 | (16.4%) |

**Table 2. Absolute and relative frequency of TMD diagnosis according to RDC/TMD axis I.**

| Variable | | n | (%) |
|---|---|---|---|
| **TMD** | Yes | 34 | (46.6) |
| | No | 39 | (53.4) |
| **Group I (GI)** | IA | 17 | (23.3) |
| | IB | 02 | (2.7) |
| **Group II (GII)** | IIA (R) | 07 | (9.6) |
| | IIB (R) | 00 | (0.0) |
| | IIC (R) | 01 | (1.4) |
| | IIA (L) | 08 | (11.0) |
| | IIB (L) | 01 | (1.4) |
| | IIC (L) | 02 | (2.7) |
| **Group III (GIII)** | IIIA (R) | 13 | (17.8) |
| | IIIB (R) | 02 | (2.7) |
| | IIIC (R) | 04 | (5.5) |
| | IIIA (L) | 07 | (9.6) |
| | IIIB (L) | 04 | (5.5) |
| | IIIC (L) | 03 | (4.1) |
| **Multiple** | GI+GII | 01 | (1.4) |
| | GI+GIII | 03 | (4.1) |
| | GII+GIII | 04 | (5.5) |
| | GI+GII+GIII | 05 | (6.8) |

R, right; L, left.

The analysis in Table 3 represents the values of inferential statistics when the main outcome data crosses with the other variables. It was observed that patients with horizontal overjet measurements less than 7 mm had less TMD (35.4%). In this case, patients who had a horizontal overjet greater than 7 mm had a higher occurrence of TMD with a statistically significant difference ($p = 0.017$).

## Discussion

This work sought to evaluate the prevalence of TMD in individuals with Class II skeletal dentofacial deformity who were referred to orthognathic surgery and associate it with sociodemographic and psychosocial characteristics. As for sociodemographic characteristics, studies on the prevalence of TMD report that the average age of the most symptomatic patients varies between 20–35 years of age, with a higher prevalence of females, as they are the main group that most seeks health services [20–22], this group also has more severe and frequent episodes of pain compared to men [7], so they tend to be more frequently diagnosed. This was also evidenced in the present study, the sample of which was mostly women 18–21 years of age. As in this study, others studies evaluating white and black patients also point to a higher prevalence of TMD signs and symptoms in white patients [23].

Studies that followed patients for up to 20 years failed to observe any cause and effect relationship between orthodontic treatment and TMD [24–26], placing orthodontics as a confounding variable, as it could be treated as both a therapy and a predisposing factor and /or initiator [24], albeit without scientific evidence to support either hypotheses [26]. However, more recent studies have shown that the correction of malocclusions could ease the TMD condition. Moreover, dentoskeletal deformities can determine myofunctional changes such as

**Table 3. Frequencies, Chi-square, *p*-Value, PR and respective confidence intervals of the outcome "Presence of TMD" associated with independent variables.**

| VARIABLE | | TMD | NO TMD | Chi$^2$ | *p*-Value | PR | CI (95%) |
|---|---|---|---|---|---|---|---|
| | | n (%) | n (%) | | | | |
| **Sex** | Men | 3 (23.1) | 10 (78.9) | 3.462 | 0.063 | 0.281 | 0.070–1.122 |
| | Women | 31 (51.7) | 29 (48.3) | | | | |
| **Age** | < 27 years | 19 (48.7) | 20 (51.3) | 0.152 | 0.696 | 1.203 | 0.478–3.030 |
| | ≥ 27 years | 15 (44.1) | 19 (55.9) | | | | |
| **Color** | White | 30 (50.8) | 29 (49.2) | 2.226 | 0.136 | 2.586 | 0.729–9.181 |
| | Black and others | 4 (28.6) | 10 (71.4) | | | | |
| **Parafunction** | Present | 17 (50.0) | 17 (50.0) | 0.296 | 0.586 | 1.294 | 0.514–3.258 |
| | Absent | 17 (43.6) | 22 (56.4) | | | | |
| **Parafunction type** | Bruxism | 5 (38.5) | 8 (61.5) | 0.413 | 0.521 | 0.668 | 0.196–2.278 |
| | Tightness | 6 (60.0) | 4 (40.0) | 0.828 | 0.363 | 1.875 | 0.428–7.300 |
| | Onychophagy | 9 (52.9) | 8 (47.1) | 0.001 | 0.978 | 1.096 | 0.622–1.931 |
| | Object interposition | 6 (50.0) | 6 (50.0) | 0.359 | 0.551 | 1.395 | 0.470–4.067 |
| **Chewing pattern** | Unilateral | 24 (41.4) | 34 (58.6) | 3.021 | 0.082 | 0.353 | 0.107–1.165 |
| | Bilateral | 10 (66.7) | 5 (33.43) | | | | |
| **Predominant side** | Left | 11(35.5) | 20(64.5) | 0.938 | 0.333 | 0.592 | 0.206–1.700 |
| | Right | 10(66.7) | 14(51.9) | | | | |
| **Masticatory quality** | Good | 8 (33.3) | 16 (66.7) | 2.485 | 0.115 | 0.442 | 0.160–1.223 |
| | Bad | 26 (53.1) | 23 (46.9) | | | | |
| **Orthodontic treatment** | Yes | 22 (44.0) | 28 (56.0) | 0.417 | 0.518 | 0.720 | 0.267–1.939 |
| | No | 12 (52.2) | 11 (47.8) | | | | |
| **Orthodontics time** | < 24 months | 24 (53.3) | 21 (46.7) | 2.124 | 0.145 | 2.057 | 0.780–5.426 |
| | ≥ 24 months | 10 (35.7) | 18 (64.3) | | | | |
| **Chin-neck** | < 40mm | 21 (56.8) | 16(43.2) | 3.083 | 0.079 | 2.322 | 0.906–5.951 |
| | ≥ 40mm | 13 (36.1) | 23(64.3) | | | | |
| **Mandibular midline deviation** | < 1mm | 20 (43.5) | 26(56.5) | 0.473 | 0.492 | 0.714 | 0.275–1.854 |
| | ≥ 1mm | 14 (51.9) | 13(48.1) | | | | |
| **Horizontal overjet** | < 7mm | 17 (36.2) | 30(63.8) | 5.663 | 0.017* | 0.300 | 0.110–0.818 |
| | ≥ 7mm | 17 (65.4) | 9(34.6) | | | | |

TMD, Temporomandibular Disorder; Chi$^2$, Chi-square test; PR, Prevalence Ratio; CI, Confidence Interval.

*p < 0,05

tongue and lip posture disorders, speech problems, swallowing and deviations in masticatory function, which are visible in up to 64% of patients with TMD [5, 27]. In addition, these changes can trigger an adaptive response that reports to overloads and a positive diagnosis for TMD, also causing masticatory damage [27]. In this study, two-thirds of the patients considered their masticatory function to be bad, which confirms the findings already mentioned.

It is well known that unilateral chewing has a side of responsibility in TMD, because it causes an excessive compression of the condylar surface on the opposite side of mastication [26, 27]. Despite that, the variable presence of unilateral chewing, showed that more than two-thirds of the patients who reported bilateral chewing also had more TMD, but without a statistically significant difference, which was not confirmed with the findings in other studies. This could be attributed to the fact that the quality of mastication was data collected through self-assessment, contrary to the studies cited that used specific diagnostic instruments, thus, this data could therefore just be an information bias.

When analyzing the association between the presence of TMD and the upper horizontal overjet greater than 7mm, we can see a positive relationship with significant statistical value. It can therefore be inferred that there is a higher prevalence of TMD among patients with a horizontal overjet greater than 7mm.

Half of the patients in the final sample with Class II dentoskeletal deformity presented a predominance of Group I, followed by Group III and Group II. Other research [9] reported a predominance of Group I TMD in relation to Group III and Group II, respectively. This same similarity can be found when analyzing the subgroups, showing that the methodological instrument is efficient for categorizing the condition and the data presented serve as a parameter for diagnostic measures of registered populations [5, 10, 28].

Despite the findings in this research it is important to emphasize some limitations of this study, such as some topics of the research that were answered by self-assessment, like the masticatory quality and the presence of unilateral chewing. In this case there was a risk of bias because the patients can often mis-estimate their abilities or parafunctions, such as unilateral chewing. Furthermore, some patients were undergoing orthodontic treatment while others were not, which can create bias in the result findings as we mentioned that the orthodontic treatment is a confounding variable in TMD.

## Conclusions

Patients with skeletal Class II dentofacial deformity who were referred for orthognathic surgery showed higher prevalence of TMD, being group of muscular disorders and degenerative disorders (i.e., Group I and III) the most prevalent diagnosis. The presence of anteroposterior discrepancy (i.e., overjet) greater than 7 mm showed a statistically significant relationship with the presence of TMD.

## Supporting information

**S1 Dataset. Data extracted during research.**
(TIF)

**S1 Table. List of abbreviations.**
(TIF)

**S1 Appendix.**
(TIF)

**S2 Appendix.**
(TIF)

## Author Contributions

**Conceptualization:** Fábio Andrey da Costa Araújo.

**Data curation:** Fernanda Souto Maior dos Santos Araújo, Tatiane Fonseca Faro, Fábio Andrey da Costa Araújo.

**Formal analysis:** Allan Vinícius Martins de-Barros, Eudes Euler de Souza Lucena.

**Investigation:** Fábio Andrey da Costa Araújo.

**Project administration:** Emanuel Dias de Oliveira e Silva.

**Supervision:** José Rodrigues Laureano Filho, Emanuel Dias de Oliveira e Silva.

**Writing – original draft:** Fábio Andrey da Costa Araújo.

**Writing – review & editing:** Thalles Moreira Suassuna, Allan Vinícius Martins de-Barros, Bárbara Araújo da Silva.

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
