## [Decision Letter · Decision Letter 0]

16 Nov 2023

PONE-D-23-34063Temporomandibular joint disorders in skeletal Class II patients referred to orthognathic surgery: A cross-sectional studyPLOS ONE

Dear Dr. da Silva,

Thank you for submitting your manuscript to PLOS ONE. After careful consideration, we feel that it has merit but does not fully meet PLOS ONE’s publication criteria as it currently stands. Therefore, we invite you to submit a revised version of the manuscript that addresses the points raised during the review process.

We look forward to receiving your revised manuscript.

Kind regards,

Martina Ferrillo

Academic Editor

PLOS ONE

Journal Requirements:

**Additional Editor Comments:**

Based on reviewers' comments, the paper should be revised before being reconsidered for publication.

Reviewers' comments:

Reviewer's Responses to Questions

**Comments to the Author**

1. Is the manuscript technically sound, and do the data support the conclusions?

Reviewer #1: Yes

Reviewer #2: Partly

2. Has the statistical analysis been performed appropriately and rigorously? 

Reviewer #1: I Don't Know

Reviewer #2: Yes

3. Have the authors made all data underlying the findings in their manuscript fully available?

Reviewer #1: Yes

Reviewer #2: Yes

4. Is the manuscript presented in an intelligible fashion and written in standard English?

Reviewer #1: Yes

Reviewer #2: No

5. Review Comments to the Author

Reviewer #1: Dears,

The paper has well-designed research methods, appropriate statistical analysis and a relatively good interpretation of the results.

-Please be sure to use only keywords accordingly to medical subject headings (Mesh word) for a better indexing.

I suggest you add a table with the list of abbreviations used in the text.

I suggest you implement the abstract in order to make it more understandable to authors.

The introduction should be expanded perhaps by adding a section on temporomandibular disorders. I recommend some references:[10.3390/jcm12072652];[10.1111/joor.13496]

The conclusion is in accordance with the objectives of the research, its results and their interpretation, as well as the relevant literature.

Regards

Reviewer #2: Dear Authors,

thank you for giving me the opportunity to revise your manuscript entitled "Temporomandibular joint disorders in skeletal Class II patients referred to orthognathic surgery: A cross-sectional study". The paper aims to assess the prevalence of TMD in subjects with skeletal Class II dentofacial deformity. he topic is interesting and in line with the journal. The paper is well written and succinct. Nevertheless, some critical issue should be addresses:

Introduction: The introduction should stress the different causes of TMD and possible linkage with other systemic disease such as vitamin D deficiency. Please, read "doi: 10.3390/jcm11216231". Moreover, I suggest, in order to have a complete framework of the disease, to add a brief overview ot the several therapeutical conservative approaches to TMD. Please, read doi: 10.1111/joor.13571.

Materials and Methods: have the patients underwent prior treatment fo TMD? please, specify this point.

Results: well done

Discussion and conclusion: Please, replace DTM with TMD at line 199. Moreover, I suggest to add the clinical implication of your findinds. The authors state that the prevalence of TMD in skeletal II is 47% versus the 65% of the general population, but in conclusion affirm that "Patients with skeletal Class II dentofacial deformity who were referred for orthognathic

238 surgery showed higher prevalence of TMD than expected for the general population". please, replace the sentence

Best Regards

6. PLOS authors have the option to publish the peer review history of their article (what does this mean?). If published, this will include your full peer review and any attached files.

Reviewer #1: No

Reviewer #2: No

---

## [Author Response · Author response to Decision Letter 0]

19 Dec 2023

All the requested changes have been made and are pointed in the document "Response to Reviewers"

---

## [Decision Letter · Decision Letter 1]

16 Jan 2024

Temporomandibular joint disorders in skeletal Class II patients referred to orthognathic surgery: A cross-sectional study

PONE-D-23-34063R1

Dear Bárbara Araújo da Silva,

We’re pleased to inform you that your manuscript has been judged scientifically suitable for publication and will be formally accepted for publication once it meets all outstanding technical requirements.

Kind regards,

Martina Ferrillo

Academic Editor

PLOS ONE

Additional Editor Comments (optional):

Based of reviewers' comments, the paper is suitable for publication.

Congrats!

Reviewers' comments:

Reviewer's Responses to Questions

**Comments to the Author**

1. If the authors have adequately addressed your comments raised in a previous round of review and you feel that this manuscript is now acceptable for publication, you may indicate that here to bypass the “Comments to the Author” section, enter your conflict of interest statement in the “Confidential to Editor” section, and submit your "Accept" recommendation.

Reviewer #1: All comments have been addressed

Reviewer #2: All comments have been addressed

2. Is the manuscript technically sound, and do the data support the conclusions?

Reviewer #1: Yes

Reviewer #2: Yes

3. Has the statistical analysis been performed appropriately and rigorously? 

Reviewer #1: Yes

Reviewer #2: Yes

4. Have the authors made all data underlying the findings in their manuscript fully available?

Reviewer #1: Yes

Reviewer #2: Yes

5. Is the manuscript presented in an intelligible fashion and written in standard English?

Reviewer #1: Yes

Reviewer #2: Yes

6. Review Comments to the Author

Reviewer #1: Dear Authors,

this paper is improved after the reviewer's recommendation. I think that this paper can be accepted

regards

Reviewer #2: (No Response)

7. PLOS authors have the option to publish the peer review history of their article (what does this mean?). If published, this will include your full peer review and any attached files.

Reviewer #1: No

Reviewer #2: No

---

## [Editor Report · Acceptance letter]

5 Feb 2024

PONE-D-23-34063R1 

PLOS ONE

Dear Dr. da Silva, 

I'm pleased to inform you that your manuscript has been deemed suitable for publication in PLOS ONE. Congratulations! Your manuscript is now being handed over to our production team.

Kind regards, 

on behalf of

Dr. Martina Ferrillo 

Academic Editor

PLOS ONE